# Effects of a Multi-Professional Intervention on Mental Health of Middle-Aged Overweight Survivors of COVID-19: A Clinical Trial

**DOI:** 10.3390/ijerph20054132

**Published:** 2023-02-25

**Authors:** Joed Jacinto Ryal, Victor Augusto Santos Perli, Déborah Cristina de Souza Marques, Ana Flávia Sordi, Marilene Ghiraldi de Souza Marques, Maria Luiza Camilo, Rute Grossi Milani, Jorge Mota, Pablo Valdés-Badilla, Braulio Henrique Magnani Branco

**Affiliations:** 1Postgraduate Program in Health Promotion, Cesumar University, Maringa 87050-390, Brazil; 2Interdisciplinary Laboratory of Intervention in Health Promotion, Cesumar Institute of Science, Technology and Innovation, Maringa 87050-390, Brazil; 3Medicine Course, Department of Health Sciences, Cesumar University, Maringa 87050-390, Brazil; 4Research Centre of Physical Activity, Health, and Leisure, Laboratory for Integrative and Translational Research in Population Health (ITR), Faculty of Sports, University of Porto, 4200-450 Porto, Portugal; 5Department of Physical Activity Sciences, Faculty of Education Sciences, Universidad Catolica del Maule, Talca 3530-000, Chile; 6Sports Coach Career, School of Education, Universidad Viña del Mar, Viña del Mar 2520-000, Chile

**Keywords:** coronavirus, health impact assessment, psychosocial intervention, rehabilitation research

## Abstract

The present study aimed to investigate the effects of a multi-professional intervention model on the mental health of middle-aged, overweight survivors of COVID-19. A clinical trial study with parallel groups and repeated measures was conducted. For eight weeks, multi-professional interventions were conducted (psychoeducation, nutritional intervention, and physical exercises). One hundred and thirty-five overweight or obese patients aged 46.46 ± 12.77 years were distributed into four experimental groups: mild, moderate, severe COVID, and control group. The instruments were used: mental health continuum-MHC, revised impact scale–IES-r, generalized anxiety disorder-GAD-7, and Patient health questionnaire PHQ-9, before and after eight weeks. The main results indicated only a time effect, with a significant increase in global MHC scores, emotional well-being, social well-being, and psychological well-being, as well as detected a significant reduction in global IES-R scores, intrusion, avoidance, and hyperarousal, in addition to a reduction in GAD-7 and PHQ-9 scores (*p* < 0.05). In conclusion, it was possible to identify those psychoeducational interventions that effectively reduced anxiety, depression, and post-traumatic stress symptoms in post-COVID-19 patients, regardless of symptomatology, in addition to the control group. However, moderate and severe post-COVID-19 patients need to be monitored continuously since the results of these groups did not follow the response pattern of the mild and control groups.

## 1. Introduction

A high prevalence of people infected with COVID-19 had psychoemotional sequelae, such as anxiety (42%), depression (31%), post-traumatic stress disorder (PTSD; 28%), and insomnia (40%) after one month of hospital discharge infection [1]. The persistent symptoms of COVID were defined as long COVID or post-COVID-19 syndrome, characterized by people with sequelae, even after medical discharge [2]. Because of this, Xiang et al. [3] point out that the COVID-19 pandemic is closely associated with a significant increase in anxiety and depressive symptoms. On the other hand, the meta-analysis by Deng et al. [4] found that the prevalence of depression in post-COVID-19 patients was 45%, anxiety was 47%, and sleep disturbances impacted 34% of those evaluated.

A cross-sectional study aimed to verify possible factors associated with depression, anxiety, and PTSD in 898 people after COVID-19, with almost one-third of them having symptoms of depression (43%), anxiety (45%), and PTSD (32%), and symptoms associated with loneliness and low-stress tolerance [5]. On the other hand, people with more resilience and greater family support showed more resistance to these symptoms [5]. Undoubtedly, looking at these people who still have psychological sequelae is essential since the deleterious conditions related to mental health negatively impact the population’s health and quality of life [6]. In addition, another epidemiological study that evaluated 1733 people pointed out that after six months of discharge from COVID-19, sequelae were still verified, such as fatigue (63%), difficulty sleeping (26%), depression (23%), and anxiety (23%) [7].

Patients with overweight and especially, obese classified by body mass index (BMI) present increased risks of moderate and severe physical symptoms of COVID-19 [8] and also, could initiate mental health problems due to a long time for rehabilitation. Given this, psychoeducation added to other intervention tools may be effective in mitigating the effects on mental health during the COVID-19 pandemic [9] through psychoeducation, mindfulness exercises, promoting social interactions, stimulating wellness and brain health, validating emotional responses, and exploring patients’ strengths and how to organize rehabilitation goals. The above-mentioned strategies are considered a tool to help reduce anxiety and depressive symptoms and promote mental health education [10]. Considering the listed aspects, this population requires greater assistance to recover mental health and quality of life after COVID-19. Therefore, the present study aimed to investigate the effects of a multi-professional intervention model on the mental health of middle-aged, overweight survivors of COVID-19. Based on previous studies [7,9], it is presumed that psychoeducation can improve COVID-19 survivors’ mental health, providing a better quality of life for these persons.

## 2. Materials and Methods

### 2.1. Study Design

This study presents an experimental design (controlled trial) of repeated measures and four parallel groups: three intervention groups (mild, moderate, and severe), and a control group (without a positive diagnosis of COVID-19). This study followed the guidelines of the Consolidated Standards of Reporting Trials (CONSORT) [11]. Multi-professional interventions conducted by exercise physiologists (physical exercise), nutritionists (nutritional intervention), and psychologists (psychoeducation) were carried out, all in groups, over eight weeks.

### 2.2. Participants

Participants were recruited via the Municipal Secretary of Health of Maringa and the Municipal Hospital of Maringa. Thus, 141 participants of both sexes were eligible for the study. It was accepted by people with the following characteristics: (i) male and female aged between 19 and 65 years old; (ii) present a positive diagnosis for COVID-19 by laboratory confirmation; (iii) having received the first dose of the COVID-19 vaccine; (iv) being overweight or obese according to the cut-off points established by the World Health Organization [12]; (v) having participated in at least 85% of the interventions [11]; (vi) participate in the pre-participation assessment; and (vii) have contracted COVID-19 between 3 January 2021 and 1 July 2021. As exclusion criteria, the following were not accepted: (i) patients with debilitating neurological diseases (i.e., Alzheimer’s, Parkinson’s disease, and plegies); (ii) reduction in intellectual capacity via completion of the cognitive failures questionnaire [13]; (iii) use of corticosteroids and/or having a chronic or acute disease that would contraindicate physical exercise; (iv) pregnancy; and (v) not signing the informed consent form. Information was obtained via screening performed in the study by Lemos et al. [8]. Preliminarily, the sample calculation was performed via G*Power software version 3.1, using an analysis of the variance of repeated measures. An effect size of F = 0.4 was estimated considering an α = 0.05 and a correlation between repeated measures of 0.5; a sample of 56 individuals was estimated for β = 80%. Considering a possible loss of follow-up, it was decided to recruit more than 140 participants. One hundred and forty-one participants were recruited, of whom five were excluded for declining to participate in the study and one for refusing to take an anamnesis. The participants were divided into four groups, considering the symptoms of COVID-19 or the control group, namely: control group (*n =* 29); mild, with no hospitalization (*n =* 41); moderate, with hospitalization, but without the necessity for oxygen support (*n =* 37); and severe, with hospitalization and the necessity for oxygen support, i.e., mechanical or non-mechanical oxygen supply (*n =* 28). There was a sample loss of 80 participants due to lack of time, lack of motivation, financial issues, and transport issues, and those participants did not carry out the assessments. Dropouts occurred between the sixth and eighth weeks of the intervention. Finally, 56 participants in the four experimental groups were evaluated before and after the intervention. The current study was approved by the local Ethics and Research Committee under 4,546,726 and registered in the Brazilian Clinical Trial Registration Platform (ReBEC) under registration: RBR-4mxg57b, in full compliance with the Declaration of Helsinki. Figure 1 shows the experimental design of the present study.

### 2.3. Procedures

Participants went to the university laboratory for medical clearance, with the following measurements being taken: measurement of body weight and height (subsequent calculation of BMI [12]) and completing a detailed anamnesis to identify the clinical picture and symptoms of COVID-19. Subsequently, the study participants answered the applied questionnaires (see sections below). Participants self-completed the instruments before and after eight weeks of multi-professional interventions after all explanations about the instruments used.

#### 2.3.1. Mental Health Continuum–Short Form (MHC-SF)

To assess the well-being of the participants, the MHC-SF questionnaire was used, consisting of a Likert scale (1 to 6) with questions that measure the following components of well-being: Emotional well-being (EWB), social well-being (SWB), and psychological well-being (PWB), in the experiences the last two weeks [14]. Higher scores represent better indices of well-being, and the instrument has validation for the Brazilian population [15].

#### 2.3.2. Impact of Event Scale-Revised (IES-R)

The Event Impact Scale (IES-R) is a validated questionnaire for tracking post-traumatic symptoms. [16], being validated for the Brazilian population [17]. The instrument consists of 22 questions on a Likert-type scale (0 to 3) (considering the last 7 days), in which the total score is obtained by the sum of the questions based on the evaluation criteria for PTSD from the Diagnostic and Statistical Manual of Mental Disorders (DSM-4) [18], with questions related to intrusion (In), avoidance (Av), and hyperarousal (Hy). High scores represent more intense symptoms of post-traumatic stress disorder.

#### 2.3.3. Generalized Anxiety Disorder–(GAD-7)

To track participants’ anxiety levels, the GAD-7 was used. The instrument consists of 7 items that assess how much the patient is bothered by feeling nervous, anxious, worried, restless, and irritated in the last two weeks. Questions were answered on a Likert-type scale (0–3), and scores ranged from 0 to 21, where higher scores refer to a higher degree of anxiety [19]. The GAD-7 was validated for the Brazilian population [20].

#### 2.3.4. Patient Health Questionnaire-9–(PHQ-9)

The PHQ-9 is composed of nine questions that verify the presence of each of the symptoms of an episode of depression presented in the Diagnostic and Statistical Manual of Mental Disorders [18]. The nine symptoms are depressed mood, anhedonia (loss of interest in doing activities and/or things), problems with sleeping, tiredness or lack of energy, change in appetite or weight, feelings of guilt or worthlessness, problems concentrating, feeling sluggish or restless, and suicidal thoughts [21]. The questions were answered using a Likert scale (0 to 4), in which higher scores represent a symptomatology closer to the depression episode. The instrument was validated for the Brazilian population [22].

### 2.4. Compositions of Interventions

Psychological and nutritional interventions were carried out on the premises of the university where the research was conducted. The multi-professional team was duly instructed and prepared to meet the needs of the participants. Theoretical-practical activities started with nutritional interventions or psychoeducation, followed by physical exercises (use of concurrent training).

#### 2.4.1. Physical Exercise

Physical exercises were performed twice a week, focused on improving cardiorespiratory and neuromuscular fitness (mainly to improve functional capacity), lasting approximately 60 min per session. The training plan consisted of resistance exercises focused on large muscle groups, and cardiorespiratory exercises (which were performed on a treadmill, vertical/horizontal bicycle, or rowing ergometer). Each training was assembled individually, according to the needs of the participants [23], and was conducted in groups.

#### 2.4.2. Nutritional Intervention

Nutritional interventions were based on the Food Guide for the Brazilian Population [24] and were performed once a week in groups. The central objective of the interventions was to instruct participants about the benefits of healthy eating for health, quality of life, and the reduction of risks associated with chronic noncommunicable diseases (NCDs). Each intervention lasted an average of 40 min, with one section per week. The following topics were addressed: (i) food pyramid; (ii) nutritional density of foods; (iii) macro- and micronutrients; (iv) association of food with health and quality of life; (v) nutritional composition of foods; (vi) differences between diet and light foods; (vii) means for preparing healthy food; (viii) nutritional education for health and quality of life; (ix) differences between fresh, minimally processed, processed, and ultra-processed foods; and (x) food to combat sarcopenic obesity.

#### 2.4.3. Psychoeducation

Psychoeducation was based on therapeutic interventions with the central objective of proposing a model for treating and preventing mental illnesses based on an educational character [25,26] and was conducted once a week in groups. This approach used concepts and information from psychology and other areas so that the individual could broadly understand their situation and other illnesses present in our society. In this sense, the following were discussed: (i) the importance of physical exercise for a better quality of life and mental health; (ii) anxiety in everyday life, how it can impact our daily lives and, therefore, how to face them; (iii) discussions about obesity today: demystification of beliefs, prejudices, and stereotypes associated with obesity; (iv) understanding of the role of food in the social, psychological, and physical spheres; (v) information on post-traumatic stress disorder; (vi) promotion of a healthy lifestyle; (vii) reflections on stress; (viii) reflections on depressive symptoms; (ix) reflections on insomnia and relaxation techniques; (x) reflections on denial; (xi) reflections on fear; (xii) reflections on binge eating; (xiii) reflections on a healthy lifestyle; and (xiv) reflections on behavior changes. In addition, information leaflets on the respective topics were distributed at each meeting to reinforce the interventions applied and promote support material for the participants and the community [27,28]. Digital resources were also used with expository classes dialoguing with multimedia resources. The objective of the interventions was to provide knowledge and the possibility of change about the psychological consequences of the COVID-19 pandemic, in addition to guiding the essential themes of our century and helping the participating individuals have greater knowledge in the face of mental disorders that were provoked by the process of infection and the COVID-19 pandemic. Each intervention lasted an average of 40 min. Figure 2 shows the methodological design of the present study.

### 2.5. Statistical Analysis

Asymmetry and kurtosis tests and visual inspection of histograms analyzed the distributions of numeric variables. After analyzing the distributions, numerical data were described by the mean and standard deviation (±) or median and 25–75 percentiles, depending on the data normality. Categorical data were described with absolute frequency and relative frequency. Differences in the scores of each instrument were evaluated via analysis of variance (ANOVA) of mixed measures (groups and time) to identify possible differences between groups, time, and/or interactions. If a significant difference was detected, Bonferroni’s post hoc was used. The homogeneity of the data was analyzed using the Levene test, and the residual distribution analysis was performed utilizing visual inspection of the residual graphs. When only the timing effect was found, paired Student’s *t*-tests were performed for each group, to verify the possible effects of each group intervention. Absolute deltas (∆) were also calculated by performing a one-way ANOVA between groups. The “eta square” ƞ^2^ effect size was calculated according to the classification established by Richardson [29], which is: 0.0099 [*small*], 0.0588 [*moderate*], and 0.1379 [*large*]. A significance level of 5% was established for all analyses. Statistical analyses were performed using the Statistica 12.0 software (StatSoft, Tulsa, OK, USA).

## 3. Results

The final sample consisted of 55 individuals, 36 (65.45%) male, with a mean age of 49.93 ± 13.08 years old, of whom 19 (35.19%) had a graduate degree, 19 (35.19%) had completed higher education, and 13 (24.07%) had only completed high school. Of the 55 participants, 40 (72.73%) had a spouse, 9 (16.36%) were single, and 6 (10.91%) were divorced or widowed. Table 1 presents the initial characteristics of the participants according to the assessed groups.

Figure 3 presents the Mental Health Continuum (MHC) questionnaire scores of the participants in this study before and after the multi-professional interventions.

At the beginning of the intervention, all the groups did not present significant differences among them for all questions of the MHC questionnaire (*p *> 0.05). As described in Figure 3, a timing effect was observed, with a significant increase in global MHC scores (F_3,52_ = 10.03; *p* = 0.002; *ƞ*^2^ = 0.04–small), EWB emotional well-being (F_3, 52_ = 6.69; *p* = 0.013; *ƞ*^2^ = 0.03–small), social welfare-SWB (F_3,52_ = 6.11; *p* = 0.017; *ƞ*^2^ = 0.03–small), and well-being psychological-PWB (F_3,52_ = 8.17; *p* = 0.006; *ƞ*^2^ = 0.03–small) after the interventions. An interaction effect between group and time for psychological well-being-PWB (F_3,52_ = 3.86; *p* = 0.014; *ƞ*^2^= 0.03–small) was also observed, with an increase in the scores of the control group after the interventions (*p* = 0.024). For the total MHC, there was no significant difference in the deltas (F_3.52_ = 0.74; *p* = 0.527; *ƞ*^2^ = 0.04–small) of the different experimental groups. For the MHC and EWB, there was no significant difference in the deltas (F_3.52_ = 1.25; *p* = 0.29; *ƞ*^2^ = 0.06–moderate) of the different experimental groups. For the MHC and EWB, there was no significant difference in the deltas (F_3,52_ = 0.49; *p* = 0.68; *ƞ*^2^= 0.02–small) of the different experimental groups. For the MHC and PWB, a significant difference was observed for the deltas (F_3,52_ = 4.20; *p* = 0.009; *ƞ*^2^= 0.19–large) of the different experimental groups, with the Bonferroni post hoc showing significantly higher values for the control group when compared to the moderate (*p* = 0.006). Paired *t*-tests showed only a significant difference in the PWB score with higher values after intervention in the control group (*p *< 0.05), and a significant difference in the MHC global score with higher values after intervention in the mild group (*p *< 0.05). Besides, there was a tendency for EWB and PWB (*p* = 0.07; for both comparisons) to have higher values after intervention in the mild group. There was only a significant difference in EWB and SWB scores, with higher values for the moderate group after intervention (*p *< 0.05). However, no significant differences were observed for the severe group (*p *> 0.05).

Figure 4 shows the impact event scale revised (IES-R) scores of the participants in this study before and after the multi-professional interventions.

At the beginning of the intervention, all the groups did not present significant differences among them for all questions of the IES-R questionnaire (*p *> 0.05). There was only a time effect for the IES-R (Figure 4), with a significant reduction in the global IES-R scores (F_3,52_ = 12.22; *p *< 0.001; *ƞ*^2^ = 0.05–small), intrusion (F_3,52_ = 10.75; *p* = 0.002; *ƞ*^2^ = 0.05–small), avoidance (F_3.52_ = 6.59; *p* = 0.013; *ƞ*^2^ = 0.03–small) and hyperarousal (F_3, 52_ = 13.72; *p *< 0.001; *ƞ*^2^= 0.07–moderate) after the interventions. For the total IES-R, there was no significant difference in the deltas (F_3,52_ = 0.74; *p* = 0.52; *ƞ*^2^= 0.04–small) of the different experimental groups. For intrusion, there was no significant difference in the deltas (F_3,52_ = 1.53; *p* = 0.21; *ƞ*^2^ = 0.08–moderate) of the different experimental groups. For avoidance, there was no significant difference in the deltas (F_3,52_ = 2.22; *p* = 0.09; *ƞ*^2^ = 0.11–moderate) of the different experimental groups. For hyperarousal, there was no significant difference in the deltas (F_3.52_ = 0.52; *p* = 0.66; *ƞ*^2^ = 0.02–small) of the different experimental groups. Paired *t*-tests showed only a tendency for IES-R global score (*p* = 0.08) and In score (*p* = 0.07) for the control group. For the mild group, there was a significant difference in the IES-R global score (*p* = 0.02), In score (*p* = 0.03), Av score (*p* = 0.05), and Hy score (*p* = 0.02), with lower values after the intervention. There was no significant difference in *t*-tests in all IES-R scores for the moderate group after intervention (*p *> 0.05). Finally, there was just a significant difference in the Av score (*p* = 0.01) with lower values after intervention in the severe group. There were no significant differences for other paired *t*-tests comparison among the groups (*p *> 0.05).

Figure 5 presents the GAD-7 and PHQ-9 scores of the participants in this study before and after the multi-professional interventions.

At the beginning of the intervention, all the groups did not present significant differences among them for all questions of GAD-7 and PHQ-9 questionnaires (*p *> 0.05). A time effect was observed, with a significant reduction in the scores of the GAD-7 (F_3,52_ = 31.96; *p *< 0.001; *ƞ****^2^*** = 0.14–*large*) and PHQ-9 (F_3,52_ = 18.15; *p *< 0.001; *ƞ****^2^*** = 0.07–*moderate*) after the interventions. However, no group effect or interaction was found between the responses of the GAD-7 and PHQ-9 (*p* > 0.05). For the GAD-7, there was no significant difference in the deltas (F_3,52_ = 1.11; *p* = 0.35; *ƞ****^2^*** = 0.06–*moderate*) of the different experimental groups. For the PHQ-9, there was also no significant difference in the deltas (F_3,52_ = 1.84; *p =* 0.15; *ƞ****^2^*** = 0.09–*moderate*) of the different experimental groups. Paired *t*-tests showed a significant reduction in GAD-7 scores for the control (*p* = 0.05), mild (*p* = 0.0002), moderate (*p* = 0.02), and severe groups (*p* = 0.03) after interventions. In addition, paired *t*-tests showed only a significant reduction in PHQ-9 in mild (*p* = 0.0005), and moderate groups (*p* = 0.01) after interventions.

## 4. Discussion

The present study aimed to investigate the effects of a multi-professional intervention model on the mental health of middle-aged, overweight survivors of COVID-19. The results of the present study confirmed that psychoeducation, added to multi-professional activities, was effective in significantly improving the psychological symptoms in different experimental groups with higher or lower emphasis depending on the disease severity of COVID-19, and even in the control group. Therefore, multi-professional interventions effectively improved the mental health and sleep quality of participants in the present study (regardless of the experimental group).

To date, to the authors’ knowledge, no studies have investigated multi-professional interventions using psychoeducation, nutritional intervention, and physical exercise (together—in multi-professional interventions) in individuals who survived COVID-19 with overweight or obesity. The excess fat could reduce physical fitness and extend the treatment to recover the COVID-19 survivors [8]. Therefore, these patients require special care, due to their physical condition, and with a long recovery period, they may develop mental health problems. The scientific literature indicates that psychoeducation added to physical activities and a multi-professional approach can positively influence psychological aspects, such as a well-being decrease in anxiety and depressive symptoms, and contribute to the treatment and prevention of depression, anxiety, and post-traumatic stress [23,30]. Some similar practices have already shown positive results, reducing the impacts caused by the COVID-19 pandemic and other psychiatric conditions [9,10,31]. Another recent study from our research group showed similar results in approaches with concurrent exercise and dietary reeducation in overweight or obese middle-aged females [32].

The COVID-19 pandemic has negatively influenced the well-being of the general population [33]. Social isolation, the COVID-19 pandemic, low social support, low family income, and other aspects directly influenced the population’s mental health [5,34,35]. Thus, multi-professional interventions are considered well-known tools for preventing and treating various physical and mental disorders [10,36]. As a result, there was a significant improvement in general and specific components of well-being: emotional, social, and psychological. These responses may result from the emotional, social, and psychological support that a multi-professional team provides since, in this type of study, all areas of mental health are worked on, thus promoting psychological support, development of self-esteem, and improved social interaction [37,38,39].

Multi-professional teams allow a holistic view of individuals, enabling personal, social, and psychological development [40], factors that may have directly influenced the three subscales assessed by the MHC-SF: psychological well-being, social well-being, and emotional well-being. Cacioppo et al. [41] showed that loneliness could be directly linked to cardiovascular disease, sleep deregulation, and high cortisol release. There is evidence that social isolation is directly associated with the inflammatory system [42]. The systematic review by Williams et al. [43] identified that multi-professional interventions have already been proven to reduce loneliness through physical exercises, cognitive behavioral therapy, and psychoeducation. Additionally, behavioral psychoeducation interventions were positive for combating post-COVID-19 sequelae in a previous study [8]. In the study, as mentioned above, an intervention model based on cognitive behavioral therapy was used with mindfulness and other tools to assist post-COVID-19 rehabilitation. However, the interventions took place in the online format, with a shorter time (1 to 2 weeks), compared to the present study, which was carried out for eight weeks in person. Finally, significantly higher values were verified for the delta of psychological well-being in the control group compared to hospitalized individuals. These significant differences may be related to the possible deleterious impacts of COVID-19 on the mental health of hospitalized individuals.

As verified in this study, the participants showed a significant reduction in the global score and the IES-r subscales, namely intrusion, hyperarousal, and avoidance. These scales are due to the diagnosis of PTSD [16]. The difference in the subscales between pre and post represents a significant improvement in the diagnosis of the syndrome since this questionnaire is based on the DSM-IV criteria [18]. Among the effects of the COVID-19 pandemic, the increase in depressive symptoms also stands out negatively [7,44]. An effect persisted even after the worst moments of the pandemic, a recurrent symptom of the post-COVID-19 syndrome [7,45]. Therefore, physical activities are a primordial tool in the fight against depressive symptoms [46]. Likewise, psychoeducational activities reduce these symptoms and are often used as a non-drug treatment [10,44]. With that, the present study presents similarities with the findings in the literature that observed improvements in the PHQ-9 and GAD-7 scores [34,47]. Considering the evidence that points to the need for care for these individuals who have post-COVID symptoms [9,47,48,49], a unique look at the population after the COVID-19 pandemic becomes relevant so that effective techniques can be developed for the treatment of all the symptoms of this syndrome. Prolonged symptoms affect individuals’ quality of life and well-being [50]. Given this, the indispensability of psychoeducation actions and interdisciplinary actions to improve the physical, nutritional, and psychosocial health of COVID-19 survivors is confirmed. Although significant improvements (time effect) were observed for all instruments applied in the different experimental groups, caution is recommended in interpreting the findings, since the moderate and severe groups alone did not follow, in some circumstances, the standards of the other groups (control and mild). Therefore, longer interventions are suggested for the groups that had more severe symptoms of the disease, and even follow-up analyses to identify the behavior of the groups over time and even possible relapses. Thus, early intervention strategies can again be incorporated, to recover the mental health of COVID-19 survivors.

Our study provides information regarding COVID-19 survivors, which is timely and informative data for the intervention and recovery of those patients. However, in our study, there was a significant sample loss over the eight weeks of intervention, with a possible lack of interest on the part of the population to continue with multi-professional care. This occurred due to a lack of motivation and time, and patients believed that they were already better and would not need to continue with multi-professional activities. Almost all of the patients who dropped out of the interventions were low-income people. Qualitative feedback on drop-outs was linked to financial and transport issues. During the most restrictive period of the COVID-19 pandemic, people received minimal financial assistance from the government and started treatment. When the resources were exhausted, people had to work or stay at home and save money to buy food and pay the essential expenses of their respective households. Unfortunately, in Brazil, the researchers and universities cannot pay the expenses of the patients. Thus, a big part of drop-out is linked to the Brazilian reality. In addition, it was not possible to perform an intention-to-treat analysis, as the participants did not return to the university to be reassessed. Finally, no studies have been found combining multi-professional interventions with psychoeducation in COVID-19 survivors. Thus, the present study’s findings can guide possible actions to recover the global health conditions of those who contracted COVID-19.

## 5. Conclusions

It was concluded that multi-professional interventions significantly improved general well-being, emotional well-being, social well-being, and psychological well-being indicators in middle-aged, overweight survivors of COVID-19. Complementarily, there was a significant reduction in the scores representing the symptoms of post-traumatic stress disorder, the general scale of intrusion, avoidance, and hyperarousal. Furthermore, it was also possible to conclude that the interventions effectively reduced anxiety and depressive symptoms due to the reduction in scores after the multi-professional interventions. A significant difference over time was also observed in all evaluated groups, suggesting that psychoeducation added to a multi-professional team has a great positive impact on the mental health of people with different symptoms of COVID-19 and even those who do not have the infection. Lastly, patients in the moderate and severe groups need to be monitored continuously since the results of these groups did not follow the response patterns of the mild and control groups. Thus, the monitoring process and follow-up analyses for moderate and severe groups are indispensable and urgent.

## Figures and Tables

**Figure 1 ijerph-20-04132-f001:**
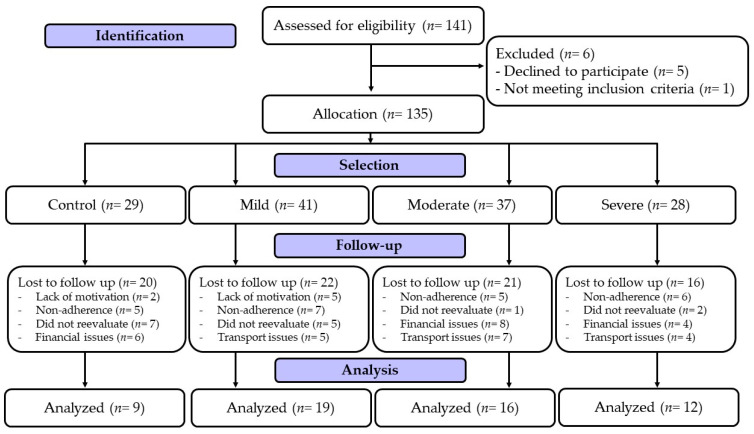
Study flowchart according to CONSORT.

**Figure 2 ijerph-20-04132-f002:**
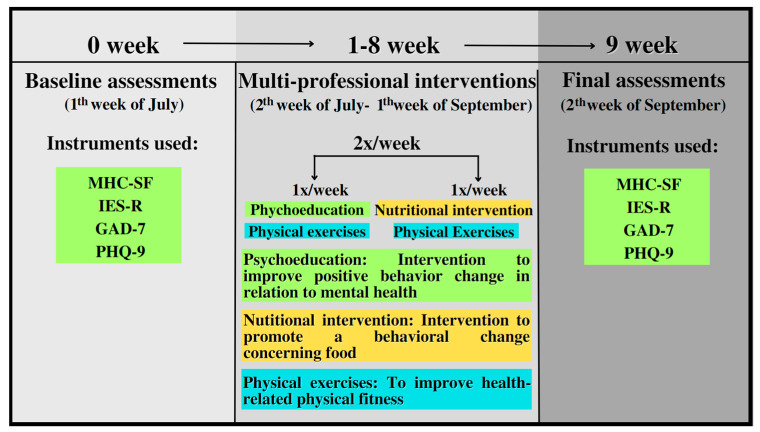
Summary of multi-professional interventions. MHC-SF = Mental Health Continuum–Short Form; IES-R = Impact of Event Scale—Revised; GAD-7 = Generalized Anxiety Disorder; PHQ-9 = Patient Health Questionnaire.

**Figure 3 ijerph-20-04132-f003:**
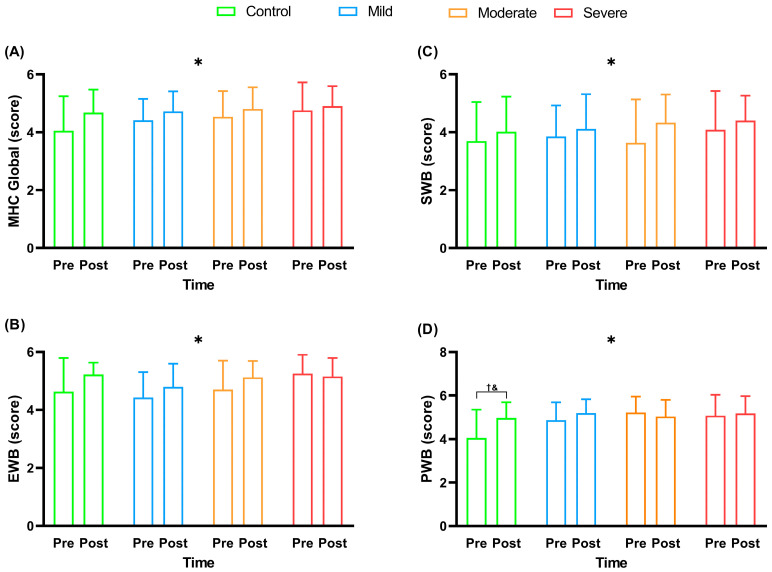
Mental Health Continuum (MHC) scores before and after multi-professional interventions. Data were described as mean and standard deviation (±); MHC = Mental Health Continuum; EWB = emotional well-being; SWB = social well-being; PWB = psychological well-being; * = timing effect (pre- vs. post-intervention); † = interaction between pre- and post-intervention for the control group (*p* < 0.05); & = interaction with significantly higher values for the control group when compared to the moderate group (*p *< 0.05); Panel (**A**) = MHC score Global; Panel (**B**) = EWB score; Panel (**C**) = SWB score; Panel (**D**) = PWB score.

**Figure 4 ijerph-20-04132-f004:**
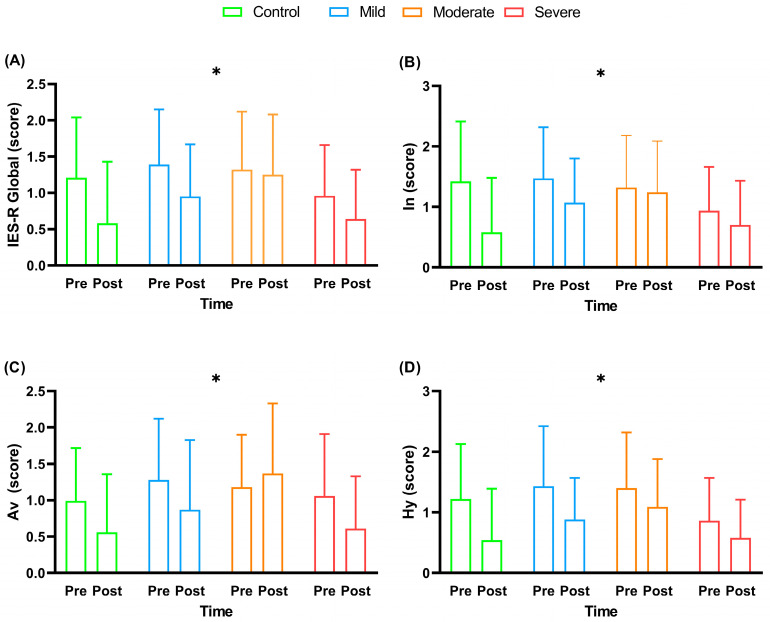
Impact event scale revised (IES-R) scores before and after multi-professional interventions. Data are presented as mean and standard deviation (±); IES-R = Impact Event Scale–Revised; In = intrusion; Av = avoidance; Hy = hyperarousal; * = timing effect (pre- and post-intervention) with *p *< 0.05; Panel (**A**) = IES-R global score; Panel (**B**) = In score; Panel (**C**) = Av score; Panel (**D**) = Hy score.

**Figure 5 ijerph-20-04132-f005:**
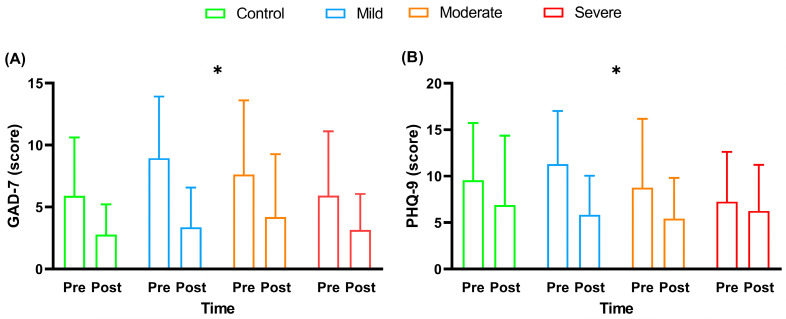
Scores of (GAD-7) and (PHQ-9), before and after multi-professional interventions. Data are presented as mean and standard deviation (±); GAD-7 = 7-item Generalized Anxiety Disorder Questionnaire; PHQ-9 = Patient Health Questionnaire-9; * = timing effect (pre- and post-intervention), with *p *< 0.05; Panel (**A**) = GAD-7 score; Panel (**B**) = PHQ-9 score.

**Table 1 ijerph-20-04132-t001:** Initial characteristics of the participants.

	Control (*n =* 9)	Mild (*n =* 19)	Moderate (*n =* 16)	Severe (*n =* 12)
**Age (years)**	45.44 ± 11.98	49.47 ± 13.46	46.63 ± 12.80	48.25 ± 10.29
**Male**	8 (88.89)	11 (57.89)	8 (50.00)	10 (83.33)
**Female**	1 (11.11)	8 (42.11)	8 (50.00)	2 (16.67)
**BMI (kg/m^2^)**	30.22 ± 5.22	28.05 ± 3.78	34.59 ± 6.83	32.40 ± 4.89
**Infirmary (d)**	-	-	10 (5.00–14.00)	14.00 (11.50–18.00)
**ICU (d)**	-	-	-	12.00 (4.00–29.00)
**Civil status**				
**Single**	3 (37.50)	4 (22.22)	2 (12.50)	0 (0.00)
**Stable union**	5 (62.50)	11 (61.11)	12 (75.00)	11 (91.67)
**Divorced**	0 (0.00)	1 (5.56)	1 (6.25)	1 (8.33)
**Widower**	0 (0.00)	2 (11.11)	1 (6.25)	0 (0.00)
**Education**				
**Postgraduate**	4 (50.00)	7 (38.89)	3 (18.75)	4 (33.33)
**University education**	4 (50.00)	5 (27.78)	8 (50.00)	3 (25.00)
**High school**	0 (0.00)	5 (27.78)	4 (25.00)	4 (33.33)
**Others ***	0 (0.00)	1 (5.56)	1 (6.25)	1 (8.33)

Note: Numeric data presented as mean and standard deviation (±) or median and 25–75 percentiles; categorical data described with absolute frequency and relative frequency (%); BMI = body mass index; d = days; * = others: elementary school.

## Data Availability

The datasets generated during and/or analyzed during the current research are available from the corresponding author upon reasonable request.

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
