# Peer review of "Effects of a Multi-Professional Intervention on Mental Health of Middle-Aged Overweight Survivors of COVID-19: A Clinical Trial"

_ijerph, 2023, doi:10.3390/ijerph20054132_

Round 1

Reviewer 1 Report

I find it an interesting and useful study, where it emphasizes the importance of a multidisciplinary intervention in post-covid overweight and obese patients.

1 It seems to me that this statement needs to be completed. Symptoms associated with loneliness and low stress tolerance……..; (line 49)

2 Detail the study by Jaywant, A. et al. (2022). And if it is possible to add other similar studies.

3 Address in the introduction the relationship between overweight and obesity and the symptoms of Covid.

METHOD

147. Include the number of sessions per week of the nutritional and psychoeducational intervention.

RESULTS

Add graphics to give more clarity to the results.

DISCUSSION

I found it well written and interesting. Considering that the participants were overweight and obese, I suggest including this aspect in this section.

Author Response

We send an answer to your comments attached.

Reviewer 2 Report

This is a review of the manuscript titled “Effects of a Multi-professional Intervention on Mental Health of Middle-aged Survivors of COVID-19: A Clinical Trial”. The manuscript presents some important psychoeducational interventions to consider in how the global post-COVID-19 treatment should be, for example, reducing anxiety, depression and PTSD symptoms. Along with this, the design and statistical analysis showed clear and evident results of the intervention in the scales applied to the sample. I find the bibliography well applied into the manuscript.

Minor revisions are needed:

1.      Title. I suggest that title needs to present a better description of the group, considering that consisted of overweight or obese patients and this is not referred to.

2.      Page 2. Participants. In which week of the program do the patients usually drop out?

3.  Figure 2. This can be improved if you include the dates of the start and end of the 8-week intervention.  

3.   Please clarify if the psychoeducational intervention was tailored according to age, civil status, and education.  Also, were the discussions about concepts and information from psychology where in group sessions or individually?

4.      In the discussion there are a couple of questions to acknowledge, 1) Considering the high rate of dropout in the groups, what could have been done to motivate them, besides economic aid?  2) What improvement in the program could motivate the patients to finish the program? 

Author Response

(The authors gave the same response as above.)
